# Calorimetry for active systems

Pritha Dolai,[1,][*] Christian Maes,[1] and Karel Netočný[2]

[1]*Instituut voor Theoretische Fysica, KU Leuven, Belgium*

[2]*Institute of Physics, Czech Academy of Sciences, Prague, Czech Republic*

(Dated: October 12, 2022)

We provide the theoretical basis of calorimetry for a class of active particles subject to thermal noise and (flashing) potentials. Simulating AC-calorimetry, we numerically evaluate the heat capacity of run-and-tumble particles in double-well and in periodic potentials, and of overdamped diffusions with a flashing potential. Low-temperature Schottky-like peaks show the role of activity and indicate shape transitions, while regimes of negative heat capacity appear at higher propulsion speeds. From there, significant increase in heat capacities of active materials may be inferred at low temperatures, as well as diagnostic tools for the activity of biological systems operating in a smaller temperature range.

## I. INTRODUCTION

The influence of heat on chemical reactions was already breaking ground in the 18-th century [1]. Yet, physics understanding remained incomplete for the specific heat of gases. That played a crucial role in the emergence of quantum mechanics and its applications to condensed matter physics. Since then, heat capacity is often depicted as a material constant, changing only by modifying volume or by varying temperature or other intensive parameters. It is however natural that activity may play a role as well, as equipartition is easily violated by driving or active forces [2]. For example, a tissue may change its heat capacity when it is acted upon by molecular motors, or a transmission device when undergoing random potential changes while maintaining relatively large heat or electric currents. Similarly and not thoroughly explored so far, heat capacity will differ between live matter and its unorganized mixture of molecules [3, 4].

---

[*]Electronic address: pritha.dolai@kuleuven.be

Quantitative explorations of heat due to biological functioning or as function of metabolic rates and changes therein are studied under the heading of bioenergetics [5]. References on measuring heat production in bacterial reservoirs include [6–8]. In general however, not much got systematized on the theory side of condensed matter and nonequilibrium statistical mechanics. The same holds true for active (meta)materials where thermal properties and functionalities may depend on nonequilibrium driving [9, 10] and it is again important to quantify the relation between heat and temperature.

The present paper takes this question of defining and computing heat capacities to the paradigmatic case of active systems [11–14]. Active matter is of growing interest for new materials and functionalities, and it appears important to scan a large temperature range for their thermal properties. In the same way, while Life processes happen on a much smaller window of temperatures, we wish to understand how heat capacities of bio-materials depend on activity parameters. Particle models obviously only shadow the complex mechanisms of Life or of active materials that cannot be sustained in thermodynamic equilibrium but, combining exact results and simulation, they are capable of highlighting important phenomena.

In the following prototypical examples feature flashing potentials and run-and-tumble particles both in periodic and under confining potentials. In all cases, except for an exact calculation in the Appendix A, we obtain the heat capacity by simulation, applying for the first time the scheme of AC-calorimetry [15–17] to active systems. We investigate the role of propulsion speed and of tumbling or flashing rates on particles that are either confined or which move on a periodic landscape. The main results are visualized in plots of the heat capacity. In the end, we also consider the heat-related (quasi-)entropy for these systems. While the models are for simplicity restricted to one dimension, we do not believe that is a serious restriction as we are not probing thermal properties near phase transitions. In fact, we are including a study of active particles in a double-well potential which imitates, in the usual mean-field sense, higher-dimensional active particle models.

For simplicity we consider here translational motion only, ignoring e.g. activity-induced vibration or rotation. The dynamical variable is a scalar, like the position on the real line without considering inertial degrees of freedom, and the irreversible work done on the particle is by the active forces or by flashing the potential.

The present paper wants to start the computation and study of heat capacities for ac-

tive matter. Thermal properties of active particles have of course been widely discussed, including [6, 8, 18–24]. Yet, heat capacities, the traditional window on "active" degrees of freedom, have not been calculated, let alone explored there. We suspect that the lack of experimental work on heat properties of active systems is mainly due to the absence so far of a theoretical framework and model calculations, a gap the present paper wants to fill.

## II. OUT-OF-EQUILIBRIUM CALORIMETRY

We refer to [15, 25, 26] for the initial theory and basic examples of nonequilibrium heat capacities $C(T)$ as function of bath temperature $T$. The idea is to estimate the quasistatic heat excess $\delta Q^{\mathrm{ex}}$ after a small temperature change $\delta T$, while holding constant a given set of system and environment parameters:

$$C(T) = \frac{\delta Q^{\mathrm{ex}}}{\delta T} \tag{1}$$

See also [27] for the general setup, including a motivation for the type of excess heat used in the present paper. In equilibrium, the excess is just the heat produced in a reversible transformation, $\delta Q^{\mathrm{ex}} = \delta Q = C(T) \, \mathrm{d}T$.

To measure or to compute (1) we apply AC-calorimetry where we vary the bath temperature at a given frequency $\omega \neq 0$, e.g. $T_t = T + \delta T \sin \omega t$. After the system relaxes (in a time which we assume is short compared with the ratio of excess heat to steady power dissipation), we measure the time-dependent heat flux $q(t)$ to find

$$q(t) = q^{(T)} + \delta T \left[ \sigma_1(\omega) \sin(\omega t) + \sigma_2(\omega) \cos(\omega t) \right] \tag{2}$$

defining $\sigma_{1,2}(\omega)$ as the in- and out-phase components of the temperature-sensitivity of the dissipation. We assume that the temperature-heat admittance decays fast enough in time. The main difference with equilibrium calorimetry is that now the DC-part $q^{(T)}$ no longer vanishes. Around a steady nonequilibrium condition, the latter provides the dominant (for $\omega \to 0$) contribution to the heat flux, whereas the heat capacity (1) becomes the next correction. Indeed, the low-frequency asymptotics of the heat current (2) is, in linear order and neglecting $O(\omega^2, (\delta T)^2)$,

$$q(t) = q^T + \delta T \left[ B(T) \sin(\omega t) - C(T) \, \omega \, \cos(\omega t) \right] \tag{3}$$

with $C(T)$ as in (1) and where $B(T)\delta T = q^{T+\delta T} - q^T$. That method, explained in [15, 28] but in essence going back to the work of Sullivan & Seidel in [16] and to [17] for equilibrium processes, is applied in each of the nonequilibrium model systems we consider below.

We emphasize that the nonequilibrium contribution cannot be reduced to a simple "thermodynamic" form; even in the quasistatic regime, the excess heat over temperature does not need and typically will not be an exact differential [29, 30], except close-to-equilibrium [31–33]. In addition, for active matter to which we turn next, there is the additional difficulty that the position-process is not autonomous (i.e., not Markovian) while the theory in [15, 25, 26] was initially developed for Markov processes.

## III. ACTIVE GAS

Consider the one-dimensional overdamped diffusive dynamics with equation of motion

$$\gamma \dot{x}_t = v\,\sigma_t - U'(\eta_t, x_t) + \sqrt{2\gamma\,T}\,\xi_t \tag{4}$$

for the position $x_t$ of effectively independent run-and-tumble particles (RTPs) with propulsion speed $v$ in a flashing potential $U(\eta_t, x)$ under standard white noise $\xi_t$. The flashing is governed by a process or protocol $\eta_t$. The prime in $U'$ denotes a derivative with respect to $x$. If moving on the circle we must have that $U$ is periodic in $x$. The ambient temperature is $T$. We are assuming here that $\gamma$ and $v$ are not depending on the temperature, which is a serious simplification. Another modeling feature is that we consider, besides the thermal noise, only dichotomous noise $\sigma_t$, not depending on temperature or location of the particle: the $\sigma_t \in \{+1, -1\}$ is flipping at fixed rate $\alpha > 0$. To set the time-scale we put friction $\gamma = 1$.

The mean energy of the particle at time $t$ is

$$E(t) = \big\langle U(\eta_t, x_t) \big\rangle \tag{5}$$

where the (process-)average at time $t$ samples the noises in (4). Locomotion changes the position to $x + \mathrm{d}x$, by which the energy changes as $U(\eta, x) \to U(\eta, x + \mathrm{d}x)$. That energy change can be decomposed into heat and work done on the particle, as says the First Law of thermodynamics. The instantaneous expected heat flux has therefore two sources: the direct

change $dU$ of the energy minus the work done on the particle. For $dU$ we apply Itô's Lemma at fixed $\eta$, $dU = (v\sigma - U'(\eta, x))U'(\eta, x) + TU''(\eta, x)$. On the other hand, the expected work done on the particle for locomotion is $w(x; \sigma, \eta) = \sigma v(-U'(\eta, x) + \sigma v)$. Therefore, the heat flux from the thermal bath to the particle is the difference,

$$
\begin{aligned}
q(x; \sigma, \eta) &= dU - w(x; \sigma, \eta) \\
&= 2v\sigma U'(\eta, x) - (U'(\eta, x))^2 + TU''(\eta, x) - v^2
\end{aligned}
\tag{6}
$$

To compute the heat capacity through the steady heat flux (3), we thus need to evaluate

$$
q(t) = \left\langle 2v\sigma_t U'(\eta_t, x_t) - U'^2(\eta_t, x_t) + T_t U''(\eta_t, x_t) - v^2 \right\rangle_t
$$

for sufficiently large $t > 0$, where also the process average $\langle \cdot \rangle_t$ uses the slowly varying temperature $T_t = T + \delta T \sin \omega t$ to replace $T$ in the strength of the thermal noise in (4).

## A. Flashing potential

To start with an exactly solvable case we take propulsion speed $v = 0$ and $U(\eta, x) = k(1 + \varepsilon\eta)x^2/2$, $0 \leq \varepsilon \leq 1$, where $\eta = \pm 1$ is flipping randomly at rate $\alpha$. We refer to Appendix A for the calculation. The stationary energy at fixed temperature $T$ is $E^{(s)} = T/2$. It depends only on temperature, giving the (false) impression that the system partitions energy in the same way as in equilibrium. Yet the heat capacity becomes nontrivial, though still temperature independent, and it reveals a fundamentally different thermal response than for a passive particle:

$$
C(T) = \frac{1}{2}\left[ 1 + \frac{\varepsilon^2(1 + 2z)}{(1 + z(1 - \varepsilon^2))^2} \right]
\tag{7}
$$

where the (dimensionless) persistence factor $z = k/(\alpha\gamma)$ appears. We thus distinguish the following regimes:

(1) $\underline{\varepsilon = 0}$: $q^{(s)} = 0$ and $C = \frac{1}{2}$ (equilibrium)

(2) $\underline{\varepsilon = 1}$: $q^{(s)} = \frac{kT}{\gamma}$ and $C = 1 + z$ ("release-and-retract")

(3) $\underline{z \to 0}$: $q^{(s)} \to \frac{kT}{\gamma}\varepsilon^2$ and $C \to \frac{1}{2}(1 + \varepsilon^2)$ (vanishing persistence)

(4) $\underline{\varepsilon < 1, z \to \infty}$: $q^{(s)} \to 0$ and $C \to \frac{1}{2}$ (infinite persistence $\alpha \downarrow 0$).

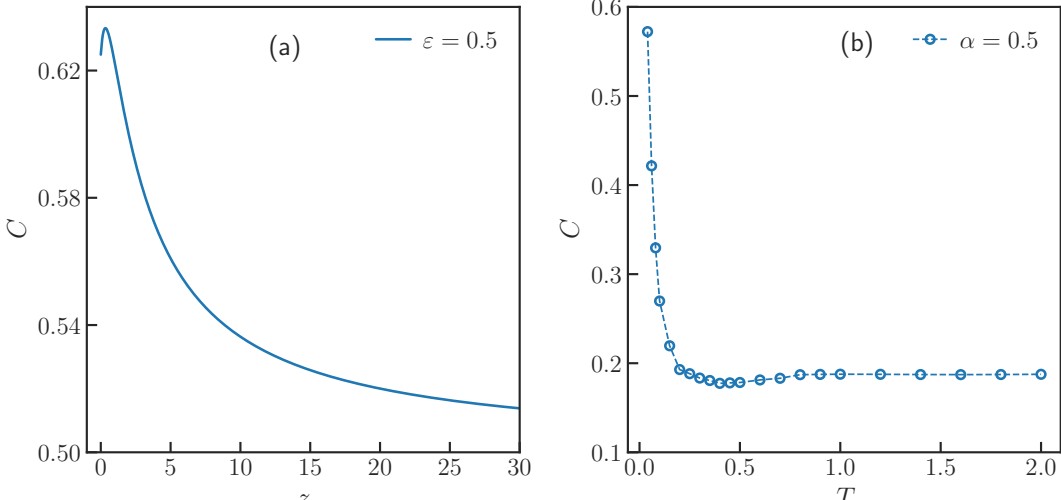

FIG. 1: (a) Heat capacity of a particle moving in a flashing harmonic potential, as in (A1), plotted as a function of persistence factor $z = k/(\alpha\gamma)$ for $\varepsilon = 0.5$. For $z \to 0$, $C \to 5/8$. (b) Heat capacity of a particle moving in a flashing symmetric double-well potential (8), plotted as a function of temperature for flashing rate $\alpha = 0.5$.

The equilibrium case is clearly separated from the active case, despite the "equipartition" $E^{(s)} = T/2$. In Fig. 1(a) we plot the heat capacity as a function of $z$ for $\varepsilon = 1/2$. We see how we can obtain the persistence factor from the heat capacity. To contrast, Fig. 1(b) gives the heat capacity for a particle in a flashing symmetric double well,

$$U(\eta, x) = \frac{k}{2}\Big(1 + \frac{\eta}{2}\Big)\Big(\frac{x^4}{4} - \frac{x^2}{2}\Big), \qquad \eta = \pm 1 \tag{8}$$

In contrast with the flashing harmonic potential, that double-well case cannot be solved exactly. Here the AC-numerical work is able to show how the (nonequilibrium) heat capacity nontrivially depends on (low) temperature.

## B. Run-and-tumble in a periodic potential

Run-and-tumble particles (RTPs) have a nonzero propulsion speed $v$ in Eq. (4). There is no flashing of the potential $U$. All the results here are obtained via AC-calorimetry, following the scheme of Eqs. (1)-(3); see also [15].

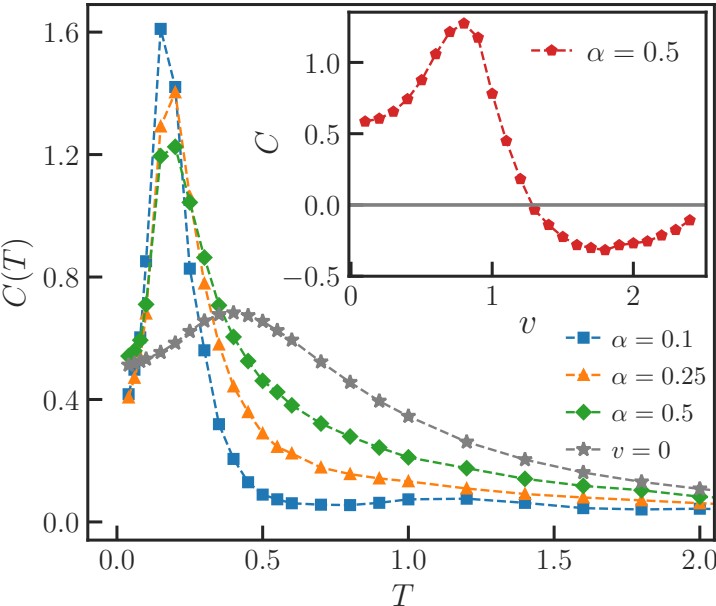

FIG. 2: Heat capacity of RTPs for different tumbling rates, moving in a sinusoidal potential with amplitude $E_0 = 1.0$ and propulsion speed $v = 1.0$. The grey line corresponds to the $v = 0$ (equilibrium) case. The inset shows the variation of the heat capacity with $v$ for fixed $E_0 = 0.5$, tumbling rate $\alpha = 0.5$ and at temperature $T = 0.1$.

Consider a RTP with position $x_t$ on the circle of length $L$, and moving in a sinusoidal potential. The dynamics is given by (4) where $U(\eta, x) = E_0 \sin(2\pi x/L)$ is not flashing. We fix $L = 20$ in the simulations.

Fig. 2 shows that heat capacity for different tumbling rates. Interestingly, $C(T)$ has a sharp peak at around $T \simeq E_0/5$, which represents a pronounced Schottky-like anomaly [34]. It indicates the presence of an energy scale $E_0$ in the potential, and the two-valuedness of the propulsion direction adds a discrete character. The peak decreases and shifts towards higher temperature for increased tumbling rate $\alpha$. The peak value grows with the persistence to yield a significant magnification of the low-temperature heat capacity ($T < 0.3E_0$) with respect to equilibrium (= grey line in Fig. 2). Of equal interest, from Fig. 2 inset, the heat capacity gets negative when the propulsion speed $v$ is large enough to reach kinetic energies above the barrier height $E_0$.

Fig. 3(a) shows the in- and out-of-phase components of the frequency-dependent heat current, defined in Eq. (2) for RTPs in a sinusoidal potential and for temperature $T = 0.3E_0$. The $\omega \downarrow 0$ of $\sigma_2(\omega)/\omega$ gives the heat capacity, as follows from AC-calorimetry.

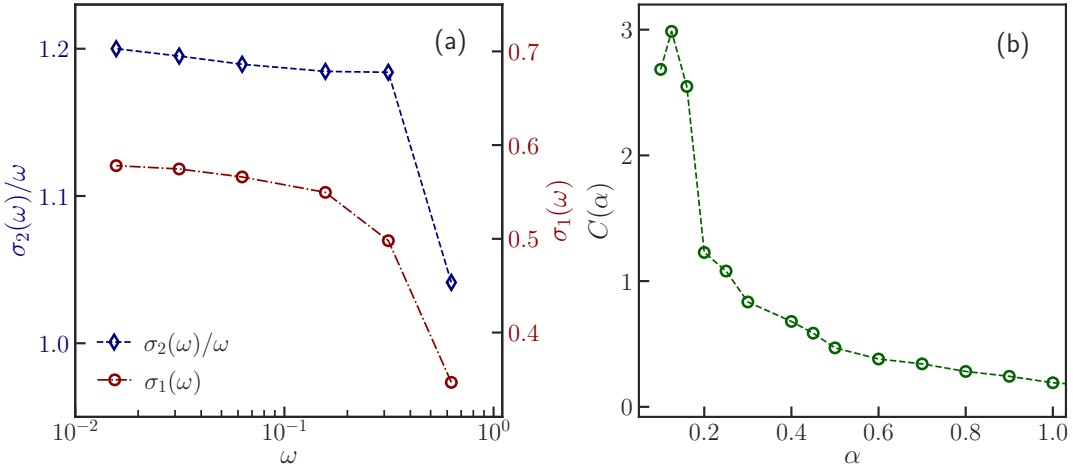

FIG. 3: RTPs in a sinusoidal potential. (a) cf. (2)-(3): the in-phase ($\sigma_1(\omega)$, as maroon circles) and out-of-phase ($\sigma_2(\omega)$, as blue diamonds) amplitudes of the heat current at $T = 0.15, \alpha = 0.5$. (b) cf. formula (9): change in excess heat per change in tumbling rate $\alpha$ at $T = 0.1$, $v = 1.0$ and $E_0 = 0.5$.

When changing the tumbling rate $\alpha$ at constant ambient temperature, there is a change in excess heat as well. We can understand it as a nonequilibrium latent heat $C(\alpha)$, the change of the excess heat per tumbling rate. As in (3), we obtain it by applying a sinusoidal modulation $\alpha(t) = \alpha + \sin(\omega t)\, \delta\alpha$ for which the heat current becomes

$$q(t) = q^\alpha - [B(\alpha)\sin(\omega t) - C(\alpha)\,\omega\cos(\omega t)]\,\delta\alpha + \mathcal{O}(\omega^2) \tag{9}$$

Fig. 3(b) depicts $C(\alpha)$ for $T = E_0/5$, showing a sharp increase for smaller tumbling rates $\alpha$, again for RTPs in the sinusoidal potential.

## C. Run-and-tumble in a double-well potential

Run-and-tumble particles in one-dimension and subject to an external confining potential have been studied for their static and dynamical properties; see e.g. [35, 36] and [37] for the

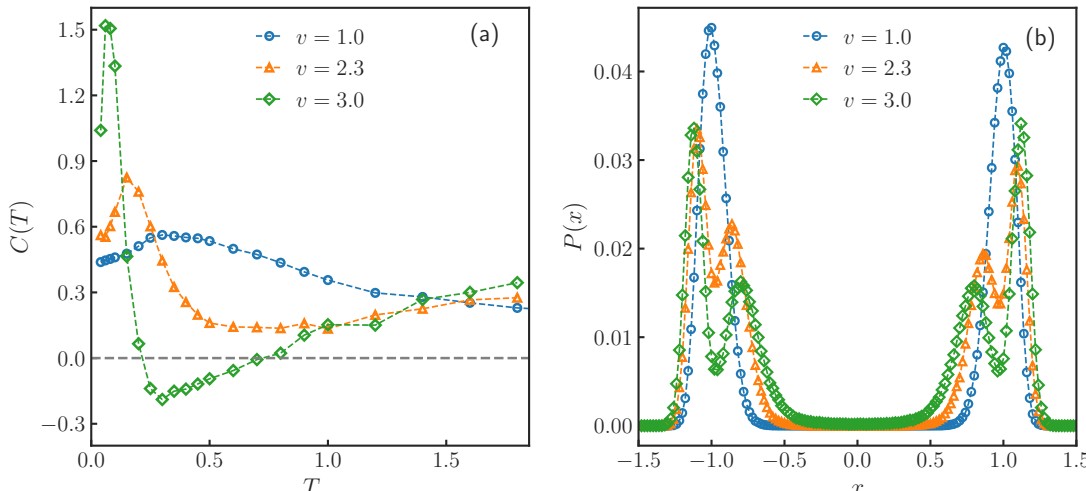

FIG. 4: (a) Heat capacity for RTPs in a double-well potential with barrier height $\Delta = 2.5$ for different propulsion speeds $v$ at tumbling rate $\alpha = 0.5$. (b) Corresponding steady state occupation density $P(x)$ for different propulsion speeds. Here $E_0 = 10.0$, and $T = 0.1$.

non-Boltzmann stationary distribution at zero temperature. Heat capacities have never been computed however. Here we consider RTPs confined by the symmetric double-well potential $U(\eta, x) = U(x) = E_0 \left(x^4/4 - x^2/2\right)$ in (4) (not flashing). The barrier height between the two wells is $\Delta = E_0/4$. As in equilibrium, that potential may arise as effective interaction in a mean-field description and may thus depend on the density profile as well, relevant for, e.g., higher-dimensional motility-induced phase transitions [38].

Fig. 4(a) gives the heat capacity for different propulsion speeds $v$. Again we see the appearance of negative heat capacities when $v$ gets large compared to the barrier height $\Delta$. Interestingly, we can also detect the zero-temperature shape transition, [35, 36, 39], from the behavior of the heat capacity at low temperature: Fig. 4(b) shows the stationary distribution and how it changes at $T = 0.1$ for the same parameters as in Fig. 4(a). When $v$ is still small, the occupation is bimodal at low temperatures (corresponding to the two wells, as in equilibrium). As we increase $v$ (going active), there appears a bimodality in each well, leading to four local maxima in the occupation distribution. That gets combined with a sharper and higher low-temperature peak in the heat capacity. And again negative heat capacities appear at large $v$ as was the case in Fig. 2 inset.

Fig. 5 gives the heat capacity of RTPs in a double-well potential for a broad range of tumbling rates $\alpha$. A broader peak shows at a temperature $T \simeq E_0/2$ for $\alpha \leq 0.5$. Compare with Fig. 6 for the small $\alpha-$regime for large propulsion speed $v$.

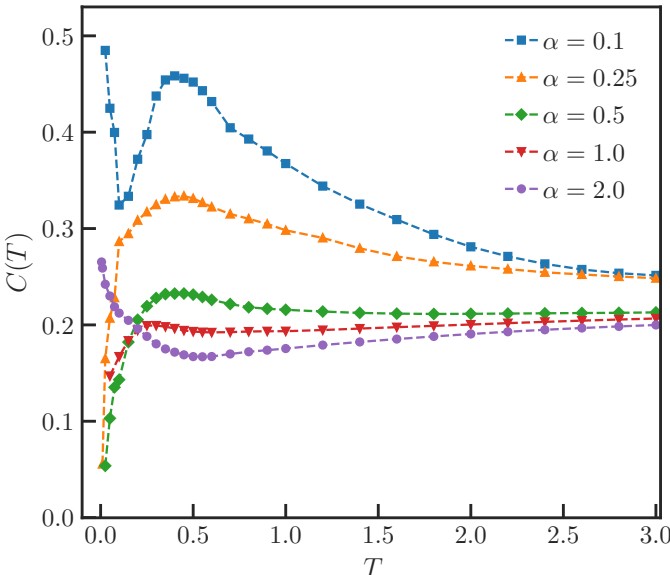

FIG. 5: Heat capacity of RTPs confined by a double-well potential plotted for different tumbling rates at $v = 1.0$ and barrier height $\Delta = 0.25$.

Finally, Fig. 6 shows a comparison between equilibrium and highly persistent RTPs. The upper curve is the heat capacity of an asymmetric (by $v$) double–well potential (aDW),

$$U(x) = E_0\big(\frac{x^4}{4} - \frac{x^2}{2}\big) + vx \tag{10}$$

and the lower curve has $v = 0$ (DW); see also [40]. For the two upper curves, the propulsion speed $v$ is large so that $U(x, \pm)$ has a single minimum at a certain $\pm x^*(v)$. For low temperatures and small tumbling rate $\alpha$, the dynamics is quickly relaxing to the neighborhood of the potential minimum $\pm x^*$ and we find that the heat capacity of the RTPs resembles the DW-curve representing Gaussian fluctuations in an effective quadratic potential, $C \sim 1/2$. When temperature increases, the heat capacity qualitatively picks up the behavior of the aDW-curve as the passive fluctuations start to be governed by the subquadratic segment in (10). At high-T, we get fluctuations in an effective quartic potential, $C \sim 1/4$.

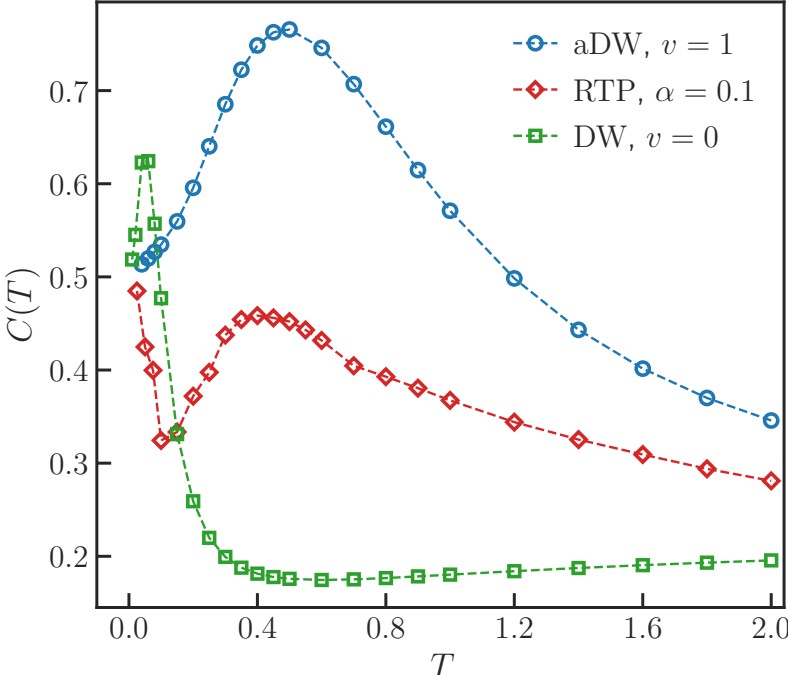

FIG. 6: Heat capacities for particles in an asymmetric double well (upper curve), for RTPs in a double-well potential at $v = 1.0$, barrier height $\Delta = 0.25$ and $\alpha = 0.1$, and for particles in a symmetric double well (lower curve). See around (10).

## D. Entropy

Entropy originates in the Clausius heat theorem, gets a statistical meaning as measure of phase volume, and gives rise to statistical forces. Such a protean entropy does not exist for genuine nonequilibria, [29, 30]. Yet, a heat-related entropy can be constructed, also for active systems, by defining the change $\cancel{\Delta}S$ in nonequilibrium *entropy*, without defining entropy as a state function:

$$\cancel{\Delta}S(T) = \int_{T_0}^{T} \mathrm{d}T' \, \frac{C(T')}{T'} \tag{11}$$

for some reference (initial) temperature $T_0$. We plot that change in Fig. 7 for RTPs in two different landscapes and for two tumbling rates. We only used the data for $C(T)$ (in Fig. 2 and Fig. 5) for temperatures $T > T_0 = 0.04$ and 0.025 respectively. As far as we know, those are the first plots of the nonequilibrium quasi-entropy (11) for active systems.

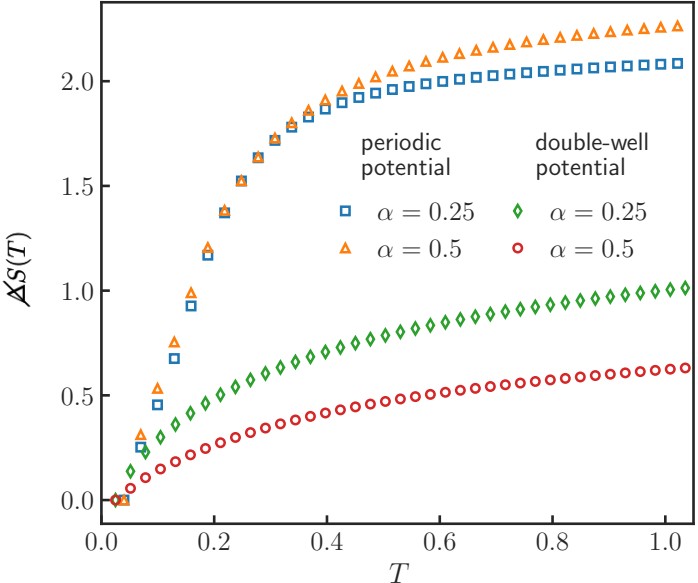

FIG. 7: Quasi-entropy (11) as a function of temperature for RTPs moving in a sinusoidal potential (upper curves), and confined by a double-well potential (lower curves). We emphasize that (11) refers to a change in temperature only.

## IV.  CONCLUSION

(Thermal) active systems are in physical contact with (at least) two reservoirs: one which is often chemical or radiative and source of low entropy, and one which can be identified with a thermal bath or environment in which energy gets dissipated. Perturbing the temperature, the heat capacity measures the excess heat in addition to the steady ever-existing dissipation. This paper has indicated how it may depend on activity parameters. For the first time for active systems, we have observed numerically Schottky-like anomalies and a regime of negative heat-capacity where an increased environment temperature enhances the excess dissipation. We are confident that low-temperature active materials show thermal characteristics as in the discussed model systems, to become a new and fascinating subject of investigation in materials science. Also for bio-systems, in a more restricted range of temperatures, the results indicate how heat capacity can serve as a diagnostic tool for activity.

Bio-physical experiments include [6, 7] but using AC-calorimetry as numerically pioneered here for active systems, would be very welcome to carry that program, to verify those

predictions and hence to continue the old adage that *Even fire is ruled by numbers* [41] in the physics of active and living materials as well.

---

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

**Appendix A: Flashing potential**

We derive the results of Section III A, to consider an overdamped diffusive particle in a one-dimensional flashing potential, following Eq.(4) with propulsion speed $v = 0$. An exactly solvable case is obtained for a quadratic potential,

$$\gamma \dot{x}_t = -\partial_x U(\eta_t, x_t) + \sqrt{2\gamma T}\, \xi_t\,, \qquad U(\eta, x) = \frac{k}{2}(1 + \varepsilon\eta)x^2 \tag{A1}$$

where $\eta_t$ is a standard dichotomous process ($\eta = \pm 1$) with persistence rate $\alpha$. The parameter $\varepsilon \in [0, 1]$ prescribes the ratio by which the potential is turned off. For understanding the First Law, Eqs. (5)-(6) in the main text, we take the $x-$component of the backward generator $L$, acting on function $f$ as

$$L^{(x)}f = \gamma^{-1}(-U'\partial_x + T\partial_x^2)f = \gamma^{-1}[-k(1 + \varepsilon\eta)x\, \partial_x + T\partial_x^2)]f \tag{A2}$$

The full backward generator is $Lf(x, \eta) = L^{(x)}f(x, \eta) + \alpha[f(x, -\eta) - f(x, \eta)]$. Therefore,

$$\frac{\mathrm{d}}{\mathrm{d}t}\langle x^2\rangle = -2k\langle x^2\rangle - 2k\varepsilon\langle \eta x^2\rangle + 2T$$

$$\frac{\mathrm{d}}{\mathrm{d}t}\langle \eta x^2\rangle = -2k\varepsilon\langle x^2\rangle - 2(k + \alpha\gamma)\langle \eta x^2\rangle$$

and under stationarity,

$$\langle x^2\rangle^{(s)} = \frac{1 + z}{1 + z(1 - \varepsilon^2)}\frac{T}{k} \tag{A3}$$

$$\langle \eta x^2\rangle^{(s)} = -\frac{z\varepsilon}{1 + z(1 - \varepsilon^2)}\frac{T}{k} \tag{A4}$$

where $z = k/(\alpha\gamma)$, a dimensionless persistence factor. Note that for flashing rate $\alpha \downarrow 0$, the variance (A3) is the sum of the variances corresponding to $\eta = \pm 1$, while the correlation (A4) remains different from zero for $\varepsilon \neq 0$. On the other hand, the stationary energy is

$$E^{(s)} = \frac{k}{2}\left(\langle x^2\rangle^{(s)} + \langle \eta x^2\rangle^{(s)}\right) = \frac{T}{2} \tag{A5}$$

and depends only on temperature.

The (mean instantaneous) heat flux equals

$$q(x, \eta) = -L^{(x)}U = \gamma^{-1}\left[k^2(1 + \varepsilon^2)x^2 + 2\varepsilon k^2\eta x^2 - kT(1 + \varepsilon\eta)\right] \tag{A6}$$

Its stationary value

$$q^{(s)} = \langle q\rangle = \frac{kT}{\gamma}\frac{\varepsilon^2}{1 + z(1 - \varepsilon^2)} \tag{A7}$$

is the steady heat flux. The quasipotential [25, 26] is

$$V(x, \eta) = \int_0^\infty dt \left[ \langle q(X_t, \eta_t \mid X_0 = x, \eta_t = \eta) - q^{(s)} \right]$$

We can find it as the solution to the equation $(LV)(x, \eta) = q^{(s)} - q(x, \eta)$ (unique on the appropriate functional space, $\|Y\|^2 = \langle Y^2 \rangle < \infty$). Substituting the *Ansatz*

$$V = \tilde{V} - \langle \tilde{V} \rangle, \qquad \tilde{V}(x, \eta) = c\,x^2 + d\,\eta x^2 + g\,\eta \tag{A8}$$

we find the corresponding coefficients

$$c = \frac{k}{2} \frac{1 + \varepsilon^2 + z(1 - \varepsilon^2)}{1 + z(1 - \varepsilon^2)}, \quad d = \frac{k}{2} \frac{z\varepsilon(1 - \varepsilon^2)}{1 + z(1 - \varepsilon^2)}, \quad g = -\frac{T}{2} \frac{z\varepsilon}{1 + z(1 - \varepsilon^2)} \tag{A9}$$

Furthermore,

$$\langle \tilde{V} \rangle = \frac{T}{2} \left[ 1 + \frac{\varepsilon^2(1 + 2z)}{(1 + z(1 - \varepsilon^2))^2} \right] \tag{A10}$$

so that $U(\eta, x) - V(x, \eta)$ remains different from zero for $\alpha \downarrow 0$:

$$\lim_{\alpha \downarrow 0} V(x, \eta) = U(x, \eta) - \frac{T}{2} \left( 1 + \frac{\varepsilon\,\eta}{1 - \varepsilon^2} \right)$$

Finally, the steady heat capacity is independent of temperature and equals

$$\begin{aligned}
C &= -\left\langle \frac{\partial V}{\partial T} \right\rangle^{(s)} = \frac{\partial \langle \tilde{V} \rangle}{\partial T} - \left\langle \frac{\partial \tilde{V}}{\partial T} \right\rangle^{(s)} \\
&= \frac{1}{2} \left[ 1 + \frac{\varepsilon^2(1 + 2z)}{(1 + z(1 - \varepsilon^2))^2} \right]
\end{aligned} \tag{A11}$$

(Note that the last term on the first line is zero, since $\partial \tilde{V}/\partial T \propto \eta$ and $\langle \eta \rangle = 0$.)