# Peer review of "Calorimetry for active systems"

_SciPost Physics_

## Round 1 · Referee Report · Anonymous (Referee 1) · 2022-11-27

Strengths

1- the authors measure a quantity called the non-equilibrium heat capacity. Very few such measurements have been made before, so this is an interesting new direction.

2- several model systems are considered, which allows some model-independent trends to be identified.

Weaknesses

1- the motivation for defining and measuring these quantities is not very clear.

2- there is rather little discussion of the physical interpretation of the results.

Report

This is an interesting manuscript about the exchange of heat between a non-equilibrium system and its environment: this is what is meant by calorimetry. Specifically, the authors analyse a quantity that they call the non-equilibrium AC heat capacity.

The results are mostly numerical. Their main purpose seems to be implementing some theoretical ideas that have been derived previously, and investigating how they play out in simple model systems.

The sciPost general acceptance criteria are satisfied. However, I am not convinced that manuscripts satisfies any of the "expectations" for sciPost Physics. In general, I think the paper is broadly suitable for publication in a sciPost journal, but significant revisions are needed.

Some specific suggestions and criticisms:

  1. The general model to be considered is eq(4). This includes as special cases run-and-tumble particles and (so-called) flashing potentials. The paper needs a clearer motivation for what these systems represent, and why they are interesting in this context. A brief review of some relevant literature would be appropriate, either in the introduction, or when the model is introduced.

Related : the introduction seems to imply that systems with flashing potentials are "active systems". As far as I can see, a system with a flashing potential would be more naturally described as a system in a time-dependent external potential. (The term "active" is usually reserved for systems where particles inject energy locally, for example by self-propulsion.)

Also, I believe that most studies of run and tumble particles are conducted without any thermal noise in the equation of motion (presumably because thermal diffusion is assumed to be negligible with respect to self-propulsion). The authors should clarify that their run-and-tumble differs from the majority of studies on such systems.

  1. In general, there is very little discussion of the results. This needs to be expanded. For example:

2a. below eq (7), four cases are enumerated and named, but there is no discussion of the physical features of these cases. What are interesting/important properties of these cases and how do these manifest in C(T)?

2b. The introduction seems to imply that experimental measurements of C(T) would provide some interesting insight into properties of active systems. However, the authors do not discuss what insights (in any) are available from the numerical results of Sec III.B, for run-and-tumble particles. The same comment applies to Sec III.C. The authors need to add (at least) a paragraph to each of these sections, to explain what the reader is supposed to learn from the results.

2c. In some cases, negative values are found for the specific heat. This should presumably have some physical interpretation. (The oscillation in the heat current is out of phase with the temperature oscillation, even at low frequencies?) What are the physical mechanisms for this effect? Are the authors able to say anything generic about what kinds of system would exhibit negative non-equilibrium heat capacities?

The authors might want to defer some of these issues to future work. However, my opinion is that the manuscript should not be accepted until the the discussion of these points has been significantly improved.

  1. The possibility of experimental measurements is mentioned several times in the Introduction and conclusion. I am not sure how this would work in practice. Any experimental system would necessarily have many more degrees of freedom than the one-dimensional position co-ordinate considered here. I think the authors should briefly discuss how their results would change if the particle that they consider has some internal degrees of freedom. Even if the equations of motion of these internal co-ordinates are decoupled from eq(4), they can still exchange energy with the heat bath. Do the authors think that calorimetric measurements could be performed with enough precision to separate the non-equilibrium heat transfer from the equilibrium-like (but still frequency-dependent) exchange of heat between the internal degrees of freedom and the heat bath?

Smaller points:

  1. Is eq (2) supposed to be valid up to corrections at O(delta T)^2 ? It is stated explicitly for eq (3) that higher order terms have been neglected, why is there no similar statement for (2)?

  2. Comparing eqs (2) and (3), are we supposed to infer some relationship like sigma1(omega) = B + O(omega^2) sigma2(omega) = C omega + O(omega^3) If this is the case, perhaps it can be stated. (More generally, the relationship between these two equations should be explained more clearly.) Also, if B and C are functions of T (as this implies), perhaps it is better to write sigma1 and sigma2 as functions of (omega,T)?

  3. In equation (10), is the v that appears in this potential the same as the the velocity 'v' that appears in equation (4)? Also, this U is a function of x alone, but the text just below seems to refer to U(x,eta) with eta=\pm1. Perhaps the authors could just write the full equation of motion for this system, that would avoid any confusion.

---

## Round 1 · Referee Report · Anonymous (Referee 2) · 2022-11-29

Weaknesses

The authors do not sufficiently motivate their definitions of work and heat and do not compare them with those used in stochastic thermodynamics. The physical meaning of negative heat capacity is not discussed.

Report

The authors analyze response of a passive system with a flashing potential and of active systems to a periodically changing temperature and calculate heat capacities of these systems.

The paper introduces notions of heat and work that seem different from those used in standard stochastic thermodynamics and in its application to active matter systems. The authors do not sufficiently motivate their choice and do not compare/contrast it with that of stochastic thermodynamics. Classic papers by Sekimoto and the more recent EPL Perspective by Fodor and Cates are not cited. While it is possible that the authors' definitions are superior, the difference with earlier works should be discussed.

The authors find a negative heat capacity for a specific range of parameters. I was wondering what it the physical interpretation of this result?

Minor comments: 1) There are some potentially confusing phrases in the paper. For example the meaning of "maintaining relatively large heat" is unclear since we teach from introductory thermodynamics that heat is a mode of energy transfer rather than a state function. On the same page there is "heat capacity will differ between live matter and its unorganized mixture of molecules" which suggests that live matter is always organized and passive (dead) matter always isn't. 2) What is the meaning of "quasi-entropy"?

Requested changes

1) The definitions of work and heat should be motivated and compared with those used in earlier literature. 2) The physical meaning of the negative heat capacity should be discussed. 3) The paper should be carefully re-read and potentially confusing colloquial statements should be re-written.

---

## Editorial Decision

resubmitted